# Familial Predisposition to Leiomyomata: Searching for Protective Genetic Factors

**DOI:** 10.3390/biomedicines10020508

**Published:** 2022-02-21

**Authors:** Maria V. Kuznetsova, Nelly S. Sogoyan, Andrew J. Donnikov, Dmitry Y. Trofimov, Leila V. Adamyan, Natalia D. Mishina, Jekaterina Shubina, Dmitry V. Zelensky, Gennady T. Sukhikh

**Affiliations:** 1Kulakov National Medical Research Center of Obstetrics, Gynecology and Perinatology, 117997 Moscow, Russia or sogoyan.n@mail.ru (N.S.S.); donnikov@dna-technology.ru (A.J.D.); d.trofimov@dna-technology.ru (D.Y.T.); l_adamyan@oparina4.ru (L.V.A.); mis7ha@gmail.com (N.D.M.); jekaterina.shubina@gmail.com (J.S.); gtsukhikh@mail.ru (G.T.S.); 2Department of Reproductive Medicine and Surgery, Faculty of Postgraduate Education of Moscow State, University of Medicine and Dentistry, 127473 Moscow, Russia; 3Department of Medicine, Kursk State Medical University, 305000 Kursk, Russia; dmitriizelenskii@mail.ru

**Keywords:** SNP-genotyping, uterine fibroids, molecular diagnostics, familial predisposition

## Abstract

In order to determine genetic loci associated with decreasing risk of uterine leiomyomata (UL), a genome-wide association study (GWAS) was performed. We analyzed a group of patients with a family history of UL and a control group consisting of patients without uterine fibroids and a family predisposition to this pathology. Six significant single nucleotide polymorphisms were selected for PCR-genotyping of a large data set of patients with UL. All investigated loci (rs3020434, rs11742635, rs124577644, rs12637801, rs2861221, and rs17677069) demonstrated the lower frequency of minor alleles within a group of women with UL, especially in a subgroup consisting of patients with UL and a familial history of leiomyomata. We also found that the minor allele frequencies of these SNPs in our control group were higher than those across the Caucasian population in all. Based on the obtained data, an evaluation of the common risk of UL was performed. Further work will pave the way to create a specific SNP-panel and allow us to estimate a genotype-based leiomyoma incidence risk. Subsequent studies of genetic variability in a group of patients with a familial predisposition to UL will allow us to make the prediction of the development and course of the disease more individualized, as well as to give our patients personalized recommendations about individual reproductive strategies.

## 1. Introduction

Uterine leiomyoma (UL) is one of the most prevalent benign neoplasms of the female reproductive system. The prevalence of these tumors varies from 50% to 70% in women of reproductive age [1,2]. The data from older women show that the estimated cumulative incidence by age 50 is approximately 70% for whites and over 80% for women of African ethnicity [3]. Approximately 25–50% of women have clinical manifestations of uterine fibroids [4,5]. Women may develop single or multiple uterine fibroids; the prevalence of multiple UL is higher than that of a single fibroid [6]. Often, surgical intervention is the only essential treatment of UL. In Russia, for example, over 50–70% of all hysterectomies are performed for fibroids, leading to a significant healthcare burden [2]. The issue of recurrent fibroids after myomectomy still poses a considerable challenge. According to the literature, the recurrence of uterine fibroids after myomectomy reaches 11.7%, 36.1%, 52.9%, and 84.4% at one, three, five, and eight years after surgical intervention, respectively [7]. This, in its turn, leads to a need for repeat surgical treatments. Such repeat myomectomies have been conducted in 1.3–27% of patients, despite the removal of all existing fibroids at the time of initial surgery. The recurrence of uterine myoma in patients with multiple and single uterine fibroids was noted in 59% and 27% of patients, respectively. Multiple uterine myomas result in a significantly higher need for repeat surgeries as compared to patients with a single myoma (26% and 11% of patients, respectively) [8,9,10]. During the last two decades, a number of studies have indicated the significance of the genetic predisposition of leiomyomata in the development of this disease [11,12,13]. Furthermore, advanced analyses of familial predisposition, including such predispositions in twins, demonstrate the genetic nature of UL development [14,15] The *MED12* somatic mutation has been identified as the one most closely associated with the risk of development of UL. In our previous study, we identified such mutations in the exon 2 of gene *MED12* as most closely associated with the risk of the development of uterine fibroids [16]. While it has been accepted that UL patients have a genetic predisposition to the condition, it is equally understood that not all primary degree relatives of women with UL develop the disease. Theoretically, this may suggest that some women with a family history of UL may possess a protective capacity against the development of uterine fibroids. This hypothesis encouraged us to search for promising genetic markers that, depending on the degree of their expression, may demonstrate protective or promoting qualities toward the development of uterine myomas. We hope that such prospective genetic “markers” will enhance individualized counseling, reproductive planning, and, in the future, perhaps offer new minimally invasive or non-invasive prevention and therapies for women with a family history of uterine leiomyoma. To this end, we investigated the frequency of occurrence of minor and major variants of alleles of six loci localized in *ESR1*, *FBN2, CELF4, KCWMB2* genes and analyzed their occurrence in patients with a personal and a family history of this disease. The objective of the study was the identification of protective gene markers against the development of uterine myomas in patients with a family history of this disease. In order to find protective loci in the genome, we conducted a genome-wide SNP-genotyping of women with UL, with and without an associated family history, that carry the MED 12 exon 2 gene somatic mutation and compared the data with women without uterine myomas.

## 2. Materials and Methods

Our study obtained approval at the IRB of V. I. Kulakov National Medical Research Center of Obstetrics, Gynecology and Perinatology. The design of our study includes two phases. In Phase I, two cohorts of patients (one with personal and family history of uterine fibroids and the other one without uterine fibroids) underwent whole genome genotyping. The confirmation of the family history of the presence of fibroids in the relatives of the patients was either earlier treatment in the V. I. Kulakov National Medical Research Center or documents on myomectomy operations in other medical centers. In Phase II, extended cohorts underwent PCR analysis for confirmation of data obtained as a result of the Phase I investigation.

### 2.1. Patients

Our study included 255 patients who underwent laparoscopy and laparotomy at the department of Gynecological Surgery of V. I. Kulakov National Medical Research Center of Obstetrics, Gynecology and Perinatology between the years 2016–2020 and met inclusion criteria. A total of 215 patients were included as subjects; 40 patients composed controls.

### 2.2. Inclusion and Exclusion Criteria

#### 2.2.1. Inclusion Criteria

Patients who underwent myomectomy or hysterectomy for symptomatic uterine fibroids during their reproductive years (18–50 years of age), who continued to have regular menstruations, and who had a family history of uterine fibroids, were included as subjects in the first phase of the study. The extended cohort of subjects in the second phase of the study included patients with a personal and family history of uterine fibroids (subgroup 1A), patients with a personal and without a family history of UL (subgroup 1B), and subgroup 1C group of patients with an unknown family history. The inclusion criteria were the absence of pregnancy, ability to understand and sign informed consent, ability to complete questionnaires, presence of regular menses, and absence of hormonal therapy for a period of 6 months prior to surgery.

The control group was composed of women in menopause who underwent hysterectomy during their postmenopausal years (over 50 years old), whose menopausal status was confirmed with hormonal assays, and who have not had a family or personal history of uterine fibroids.

#### 2.2.2. Exclusion Criteria

The exclusion criteria included postmenopausal bleeding, malignant or pre-malignant neoplasm of the reproductive system, acute inflammatory disease, adenomyosis, endometriosis, and pregnancy.

### 2.3. Intervention

The patients in both phases of the study included laparoscopic or abdominal hysterectomy/myomectomy for the study groups and other surgeries, including surgery for ovarian tumors (cysts) and pelvic organ prolapse, for the control groups.

All cases of UL were confirmed by histological examination, and the *MED12*-status of each patient in Phase 1 was positive. Each patient provided a blood sample and completed a questionnaire on clinical, reproductive, and family history. Tissue samples of uterine myomas were obtained during myomectomy or hysterectomy in all patients in the UL group. The absence of other visible pathology was confirmed surgically. Hysteroscopy and endometrial biopsy were performed in all patients. 

### 2.4. Tissue and Blood Samples Processing

DNA was isolated from blood and tissue samples using the QIAamp DNA Mini Kit (Qiagen, Hilden, Germany), according to the manufacturer’s instructions. DNA samples passed quality control (260/280 absorbance ratio 1.80–2.0 as well as visual control; the DNA integrity used electrophoresis in 1.5% agarose gel). All tissue samples of fibroids were tested for the presence of somatic *MED12*-mutations using PCR amplification and Sanger sequencing.

### 2.5. Whole Genome Genotyping of Initial Cohorts (1st Phase of the Study)

#### 2.5.1. Microarray Assay

A whole-genome scan of single nucleotide polymorphisms was performed using Genome-Wide Human SNP 6.0 arrays (ThermoFisher, Waltham, MA, USA) containing about 906,600 SNPs. The manufacturer’s instructions were followed for the labeling of 250 ng DNA and hybridization. After staining and washing using a GeneChip Fluidics Station 450, the arrays were scanned with a 3000 7G Scanner. All obtained microarrays corresponded to quality control parameters. The whole genome SNP-genotyping lists for each DNA sample were obtained using ChAS software (ThermoFisher, Waltham, MA, USA). The minor allele frequency (MAF) was calculated with the genotype information from the dataset that consisted of patients with both a personal and family history of uterine fibroids.

#### 2.5.2. Genotypes Calling

The selection of SNP loci with a significant association with UL or healthy controls was conducted using Fisher’s exact test. Candidate SNPs for subsequent analysis were chosen in accordance with the frequency of minor alleles that was higher than 0.05 and lower than 0.2. Six selected SNPs have the lowest p-value (10^−3^–10^−5^) among all the points located on the corresponding genes.

### 2.6. PCR-Genotyping of Large Cohorts (2nd Phase of the Study)

After Phase I of the study was conducted, selected single-nucleotide polymorphisms were analyzed by the individual genotyping of a large set of samples during Phase II of the study. The extended cohort of patients in the second phase of the study included patients with a personal and family history of uterine fibroids (subgroup 1A), patients with a personal and without a family history of UL (subgroup 1B), and subgroup 1C group of patients with an unknown family history. After choosing promising SNPs, a set of specific primers was created for each polymorphism, and all samples were amplified and sequenced using ABI PRISM 3130 Genetic Analyzer (ThermoFisher, Waltham, MA, USA). Based on the obtained sequence chromatograms, the genotypes of the polymorphisms studied were determined in the BioEdit biological sequence alignment editor. 

#### Statistical Analysis

Data analysis was performed using Microsoft Excel software. For each SNP, allele and genotype frequencies were computed in UL and healthy control groups. The Pearson’s standard χ^2^-test and odds ratio (OR) with a 95% confidence interval (95%CI) was used to test the correlation of allelic frequencies of the risk for developing UL in the group of patients with leiomyomata with and without a familial history and in healthy control groups. The significance level was adjusted using the Bonferroni correction (also known as the Bonferroni type adjustment), taking into account that six variants were analyzed. The null hypothesis was rejected at *p* < 0.05/6 = 0.0083.

## 3. Results

Our study included a total of 255 patients (215 patients with UL and 40 controls), who underwent laparoscopy and laparotomy in the department of Gynecologic Surgery of National Center in Russia and met inclusion criteria. A laparoscopic approach for myomectomy and hysterectomy was used in 86% (183/215) of cases and laparotomy in 14% (32/215). Myomectomy was carried out in 91% and hysterectomy in 9% of cases; 98 patients were planning a pregnancy. During 1–3 years of our observation in the framework of this study, pregnancy occurred in 57% of women (56/98). 

Three subgroups were selected from the general group of patients with fibroids. The first one (1A) included only patients with a reliably confirmed family history of this disease. The second subgroup (1B) consisted of patients without a family history of uterine fibroid (the only case among all known close relatives), and the third subgroup (1C) was formed from patients who did not have any information about their family anamnesis. 

### 3.1. Basic Clinical Characteristics

#### 3.1.1. Whole Genome Genotyping—Phase I

During this initial phase of our investigation, 20 patients with a personal and family history of UL (study group) and 14 patients with no uterine fibroids (control group), identified both by imaging and visual inspection in surgery and confirmed histopathologically, were included in the study. All patients of our study group had somatic *MED12* mutations in surgically removed fibroids. The mean age of patients with symptomatic uterine myoma (UL group) in this phase of the investigation was 34 ± 11 years; the mean age in the control group was 60.8 ± 8.9. The most prevalent symptom was uterine bleeding, in 70% (14/20) of cases, followed by pelvic pain and dysmenorrhea (55%, 11/20), anemia (30%, 6/20), and dysfunction of the pelvic organs—disturbance of urination and defecation (10%, 2/20). The majority of the UL group patients had multiple fibroids (70%, 14/20), and only six women had a single fibroid (30%, 6/20). The mean size of the tumor was 11 ± 10 cm.

#### 3.1.2. Extended Cohorts for PCR-Analysis—Phase II

Our extended cohort investigation included 255 patients; there were 215 patients with uterine myomas, with and without a corresponding family history (extended UL group), and 40 patients in the extended control group. This extended cohort included the patients studied in the 1st phase of the investigation. Among the patients in the extended UL group, 98 women had and 94 women did not have a family history of myomas (1A and 1B extended UL subgroups); 23 patients in this group did not have information about their anamnesis and composed subgroup 1C. The mean age of all patients in the extended UL group was 33 ± 14 years of age. The ages of women with myomas and associated family history, without a family history, and with an unknown family history were 34 ± 11 years old, 30.5 ± 11.5 years old, and 38 ± 9 years old, respectively. All women were of reproductive age, and the mean age in the extended control group was 58.79 ± 7.53 years.

Abnormal uterine bleeding and menstrual irregularity in Phase II of the study were observed in 59% (128/215) cases, pelvic pain and dysmenorrhea in 19% (41/215), anemia in 43% (92/215) of patients, dysuria and the disturbance of defecation in 13% (27/215), and infertility in 7% (15/215) of patients with UL. Intramural location of myoma was observed in 35% (75/215) of patients, the subserosal location was noted in 7%, and the submucosal location was seen in 6% of patients; 52% (112/215) of all patients with UL were found to have various localization. Multiple myomas were noted in 62% (134/215) of the UL patients (65, 52, and 17 patients in subgroups 1A, 1B, and 1C, respectively). Solitary myoma was observed in 38% (81/215) of all women with uterine fibroids, including 34 women with associated family history and 41 women without it; 6 patients with solitary myoma did not have information about their familial history (*p* < 0.01). The average size of fibroids was 12.65 ± 12.35 cm in all extended UL subgroups. The uterine fibroid sizes of 0.3 to 21 cm were observed in subgroup 1A; the size of the fibroids in subgroup 1B was 0.5–25 cm and was 0.5–10.4 cm in subgroup 1C, *p* = 0.96. No related statistically significant difference was established.

Interestingly, during the observation period, spontaneous pregnancy occurred in 22% of women in the 1A group, 57%—in 1B group, and 62%—in the 1C group (*p* < 0.001), with the use of ART methods in 5% women in 1A group, 19%—in 1B group, and 16%—in 1C group (*p* = 0.037). Such a result may indicate that a burdened family history of uterine fibroids can create serious problems for the realization of the reproductive function of patients. In 15% of cases, male and tubal-peritoneal factors of infertility were detected. The other patients considered pregnancies in the near future. 

### 3.2. Allelic Frequencies Comparison

The Phase I data of this study allowed us to identify six single-nucleotide polymorphisms which are characterized by a statistically significant difference in allelic frequency between the group of women with a personal and family history of uterine fibroids compared to the control group (*p* < 0.05). We conducted a meta-analysis of 906,600 SNPs in the blood of cohorts and identified six SNPs that demonstrated a statistically significant difference in the comparison of the study group with the general population. These identified six SNPs (rs3020434, rs11742635, rs124577644, rs12637801, rs2861221, and rs17677069) that corresponded to genes *ESR1, FBN2, CELF4,* and *KCWMB2* were selected for further analysis in the Phase II study of extended cohorts and controls. For all these SNPs, the frequency of minor alleles was in the range between 0.05 and 0.2. The data about SNPs identified in Phase I of our study are shown in Table 1, which presents the positions of all SNPs based on a -10 log *p*-value. The level of frequency differences of the selected SNPs has the most statistically reliable level of *p*-value throughout each gene where each SNP is positioned/settled (Figure 1). All SNPs were located in introns. Two pairs of selected SNPs were located on the same genes, FBN2 (rs11742635; rs17677069) and CELF4 (rs12457644; rs2861221).

Table 1 also contains data about the occurrence of minor alleles of each single nucleotide polymorphism in the studied groups and the general populations of different races (1000 Gen database; https://www.internationalgenome.org/, accessed on 2 December 2020). Such significant difference in the allelic and genotypic distribution of the six selected SNPs was also detected in the extended cohorts of Phase II of our study (Table 2), demonstrating significantly higher frequencies of all studied minor alleles in the extended control group and significantly lower frequencies among all UL patients in the extended cohort (all subgroups collectively). Our further analysis revealed that the subgroup 1A patients (with both a personal and familial history of uterine fibroids) demonstrated significantly lower frequencies in minor alleles of four of the six studied SNPs (rs3020434, rs124577644, rs12637801, and rs17677069) when compared with such frequencies in the extended control group (*p* < 0.05). While rs12637801 revealed a statistically significant difference between the extended control group and all patients with uterine myoma in Phase II of the study (*p* = 0.006), it failed to demonstrate such differences specifically between the extended control group and subgroup 1B of patients with uterine fibroids who had a personal, but no family history of UL (*p* < 0.05). A statistically significant difference in the frequency of minor alleles of SNPs was established between subgroups 1A and 1B of the extended cohort of patients with uterine myomas. It was found that a statistically lower frequency of these alleles was observed in patients of subgroup 1A in which both a personal and family history of myoma were present when compared with the subgroup 1B of patients with uterine fibroids that did not have an associated family history (*p* < 0.05). We also analyzed six single-nucleotide polymorphisms (rs3020434, rs11742635, rs124577644, rs12637801, rs2861221, and rs17677069) in women with single and multiple uterine myomas. The distribution of alleles in the study groups is shown in Table 3. A statistically significant difference is observed only in the case of rs12637801 (in groups of multiple and solitary myomas), suggesting that the carrier of a homozygous variant of the CC (major) allele can potentially predict the development of multiple uterine fibroids: 83.5% of women with a single myoma had this variant of allele and 71.6% in the case of multiple myomas (*p* = 0.031). Additionally, the CC variant of rs12637801 is more often detected in women with uterine fibroids (1A, 1B, 1C groups) compared to the control group: 70% women with myoma and 50% women in the control group (*p* < 0.001). A statistically significant difference was also observed in the case of the homozygous AA variant: it was seen in 20% of the control group patients and 3%, *p* < 0.001, in the group of women with UL which, on the contrary, indicates a somewhat “protective” role of this minor variant of the allele. However, there were no statistically significant differences in subgroups 1A, 1B, and 1C either in the case of the CC variant or in the case of the AA variant of rs12637801 (*p* > 0.05).

## 4. Discussion

According to the literature, cytogenetic and molecular investigations suggest that each uterine fibroid is an independent monoclonal process that arises from the proliferation of a single cell [17]. Molecular data elucidated a presence of somatic *MED12* mutations (predominantly located in exon 2) in 70–80% of myomas [18,19,20,21]. Some studies revealed that such mutations also occur in both benign and malignant mammalian smooth muscle tumors [22]. Such somatic mutations (usually called “driver-mutations”) are widespread phenomena in different pathogenic (as well as oncogenic) processes [23,24,25,26]. Nevertheless, the exact relationship between specific genetic mutations and a disease phenotype for UL remains understudied. Additionally, our previous studies demonstrated that in cases of multiple myomas, most of the fibroids harbored various *MED12* mutations. Such *MED12*mutations were shown to be more prevalent amongst patients with uterine myomas who have a family history of this disease [17]. Unfortunately, the molecular mechanism of *MED12* mutation occurrence remains unclear. It has so far been established that the only mutation found in the uterine fibroids is a mutated variant of the *MED12* gene [27]. Other somatic mutations, including overexpression of *HMGA2* gene, occur in 20–30% of tumors and most often these are solitary fibroids [28].

According to the previously published data, maternal history of leiomyoma presents the highest risk for the development of UL. Earlier studies have indicated the strong genetic influence on UL susceptibility based on linkage, population disparity, and twin studies. Most of the loci obtained were considered as “pathogenic” or “risk alleles”. Quite often, findings indicated that some polymorphisms had a strong association with the development of UL, but after they were tested using another data set, the original association did not find a confirmation [29]. Such a result is a frequent consequence of the genetic peculiarity of the population investigated. In 2012, Eggert et al. published results of a large-scale genome-wide linkage scan (GWLS), reporting discoveries in chromosomes 10p11 and 3p21 in a study of two independent cohorts of white women (a total of 261 families with a family history of UL). The authors reported a genome-wide significant result for rs4247357, and inotropic SNP in CCDC57, from self-reported case-control investigations from the Women’s Genome Health Study and the Australian Cohort Association Study. However, the loci identified as associated with an increased risk of UL in the Japanese GWAS [30], the Australian GWLS, and the European GWAS do not have notable overlaps, considerably supporting genetic heterogeneity in UL predisposition among various ethnic groups. The two-stage case-control meta-analysis of genetic variants in women of European and African ancestry with and without fibroids identified 326 genome-wide significant variants in 11. All GWAS and GWLS mentioned in this study have the same strategy in sampling. Authors combine data from one or several biobanks fibroid cases and controls. The familial history of these participants has not been analyzed, usually due to the lack of such data. Furthermore, an additional contributor to inconsistencies in data could be due to the selection criteria of the subjects in the control groups. Controls selected are often of reproductive ages. Edwards and colleagues used cohorts in which women with a personal history of uterine fibroids were younger than the women in the control group [31]. At this age, if they are not experiencing premature ovarian failure, women still carry a potential risk for the development of uterine fibroids. Selection of women in postmenopausal age/state essentially eliminates such a possibility and allows for a more reliable comparative analysis. Based on our literature review, our study represents the first attempt to find protective alleles by using a more restrictive approach to cohort recruitment. The aim of our investigation was to identify *prospective* genetic markers for the development of uterine fibroids in patients with a family history of this disease and compare the expressions of these prospective markers with those of women with uterine fibroids/leiomyomata who do not have a family history of this disease or do not know of their family history.

Identification of such prospective markers, in our view, could lay the groundwork for the development of complex and highly reliable prospective markers for the development of uterine leiomyomata and, in the future, may help physicians to enhance the level of precision in individualized counseling for women considered to be at risk for developing uterine fibroids or for women who are not certain of their family history. These prospective markers can potentially optimize early diagnosis and suggest the degree of the risk of UL development in women of reproductive age/state. The knowledge obtained in the development of these prospective markers may possibly result in the creation of non-invasive and minimally invasive preventive measures and treatment techniques for women of reproductive age/state who have or do not have a personal or family history of this disease. Information on the presence or absence of uterine fibroids in the family history of women conceived as a result of artificial reproductive technologies with the utilization of donor gametes, or a family history of women adopted as children, may be missing. Prospective genetic markers for the development of this disease may be of special interest in this group of women. 

The statistically significant variants of SNPs that we found can be used to create genetic diagnostic panels for predicting the risks of developing myomas, especially in the case of “family forms” of this disease. These SNP-based panels may help not only to optimize early diagnosis and prediction of risks, but can consequently lead to improved reproductive outcomes, increase the number of spontaneous pregnancies, and may be used to find new therapeutic approaches and opportunities for preventing the development and recurrence of uterine fibroids.

Uterine myomas are highly prevalent in the general population; the incidence is 70–80% of the female population [2,32,33]. Recurrent myomectomy is needed in more than 20% of cases after myomectomy [2,10,34]. While myomectomy carries the highest degree of preservation of the uterine circulation amongst various methods (MRI- guided focused ultrasound technique, embolization, and myomectomy) [35,36,37], the reduction in uterine circulation is still noted [34,36] and reoperations or surgeries for multiple myomas are known to carry a higher degree of surgical and obstetrical risks. Reoperation is seen more often in patients who underwent myomectomy for multiple myomas [10,38]. Such markers may allow for an assessment of the risk of development of multiple and recurrent uterine myomas for assessment of necessary timing for pregnancy in women after myomectomy conducted for symptomatic uterine fibroids or in women desiring fertility. The study of markers for recurrent fibroids is underway in our center at this time and will be the subject of a separate investigation. This study suggests the protective nature of minor alleles of the polymorphisms we discovered. We decided to investigate our data in patients with uterine fibroids who did and did not know their family history of uterine fibroids. In this study, for the first time, we found single-nucleotide polymorphism variants that demonstrated a statistically significant difference in the frequency in women with uterine fibroids with a present, absent, or unknown family history of this disease. We also studied the relationship among these polymorphisms with the presence of multiple or single uterine fibroids in these patients. 

Analysis of the methods and timing of reproduction permitted the identification of statistically significant differences in women with and without a family history of uterine myoma. Thus, the frequency of both spontaneous pregnancy and pregnancy resulting from the use of assisted reproductive methods after myomectomy in women without a family history is higher.

Taking into account the methods and frequency of pregnancy in women with uterine fibroids with and without a family history of uterine fibroids, it is advisable to review the timing of the implementation of reproductive function. If there is a family history of uterine fibroids, patients should be recommended to undergo a genetic study for prospective markers of the development of uterine fibroids in order to optimize the timing of the realization of reproductive potential.

Accordingly, examination for the presence of a predisposition to the development of uterine fibroids in young patients who have not realized reproductive function will be of key importance in planning pregnancy (timing and methods of reproductive function realization).

Based on this research, we can create a specific SNP panel for diagnosis of the risk of the development of uterine fibroids based on germ-line prospective genetic markers. The possibilities of molecular-genetic research methods are actively used in the diagnosis of various diseases: for example, mutations in the BRCA 1 and BRCA 2 genes in the diagnosis of breast and ovarian cancers [39,40] and the analysis of the FMR1 gene in the prognosis of primary ovarian insufficiency risks [41]. Using the NGS method, which allows simultaneous analysis of multiple genes in a single panel, we can diagnose MODY diabetes by examining a number of genes, such as GCK, HNF1A, HNF4A, HNF1B, ABCC8, INS, and KCNJ11 [42].

Thus, the use of genetic panels is a promising direction in the diagnosis and prediction of the risks of many diseases, including uterine fibroids. Certainly, our results can be extrapolated only to the European population and only with some restrictions, since population frequencies can vary greatly, however, within the Russian population we found approximately the same frequencies of the studied alleles in samples collected in different regions (unpublished data).

To date, there are no genetic test systems that can be used to predict the risk of development or recurrence of UL. Based on the prospective markers we found, we plan to conduct a larger study to propose genetic test systems before they are put into practice which can optimize the management tactics of patients with this disease, as well as allow for early preclinical diagnosis and prediction of the risks of recurrence of this disease.

## 5. Conclusions

Analysis of the whole-genome screenings selected SNPs in women with uterine fibroids and a familial history of this disease revealed statistically significant differences in allele frequencies compared to the control group and the general population group. In particular, rare alleles of rs3020434, rs11742635, rs2861221, and rs17677069 polymorphisms were absent in patients with anamnesis of myomas which may indicate a “protective” role of these allele variants in the development of UL, particularly in women with a family history of myoma, and a high frequency of homozygotes alleles may indicate their involvement in the development of uterine fibroids. In the case of rs12637801, the homozygous variant of the CC allele can be a predictor of the development of multiple uterine fibroids. These results can be used to create specific genetic tests (based on the evaluation of complex genetic markers) to predict the risk of development of uterine fibroids in order to enhance individualization of reproductive planning and to consider gene-based preventive and treatment strategies.

## Figures and Tables

**Figure 1 biomedicines-10-00508-f001:**
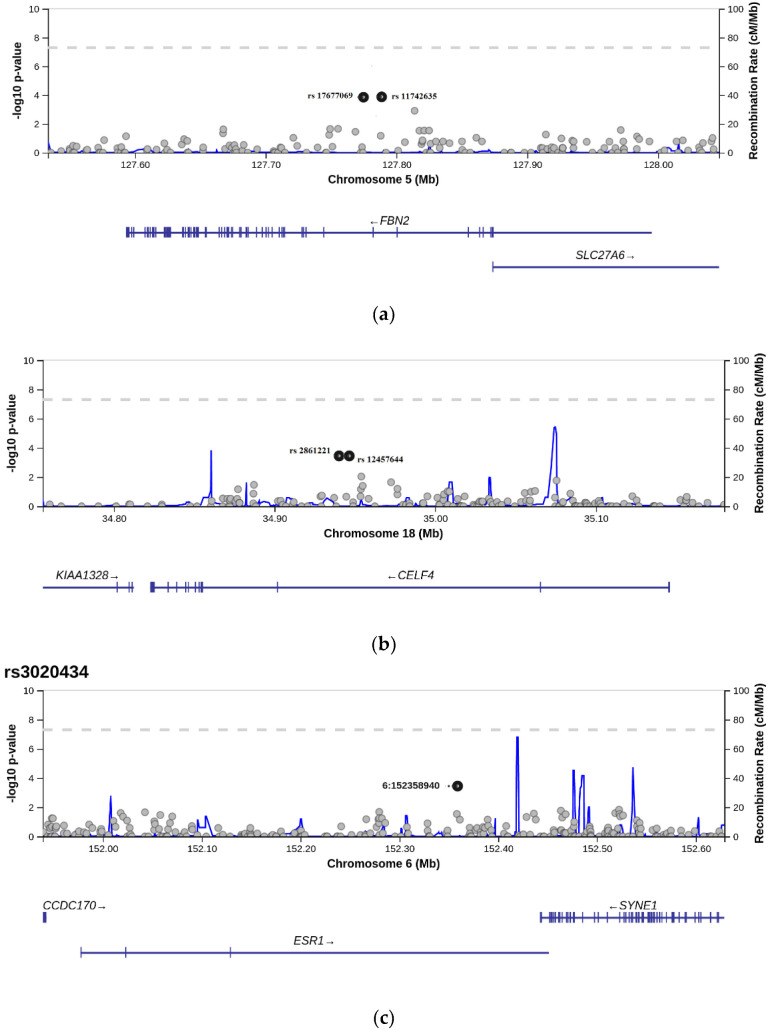
Selection of candidate single-nucleotide polymorphisms using *p*-value. The SNPs of interest are marked with a black dot. Their *p*-value reaches the maximum *p*-value. That is, the allelic differences between groups in these SNPs are the largest. (**a**) Selection of rs11742635 and rs17677069. Each point corresponds to a -log10 *p*-value of unique rs located on gene *FBN2* (*p*-value: 0.00014 for both SNPs). (**b**) Selection of rs124577644 and rs2861221. Each point corresponds to a -log10 *p*-value of unique rs located on gene *CELF4* (*p*-value: 0.00036 for both SNPs). (**c**) Selection of rs3020434 (bold point). Each point corresponds to a -log10 p-value of unique rs located on gene *ESR1* (*p*-value: 0.00036). (**d**) Selection of rs12637801 (bold point). Each point corresponds to a -log10 *p*-value of unique rs located on gene *KCNMB2* (*p*-value: 0.00038).

**Table 1 biomedicines-10-00508-t001:** SNPs selected by whole-genome analysis.

NCBI rs ID	Gene	Chromosome Location/Position	Alleles	Frequency (1000Gen)	Location
rs12637801	*KCNMB2*	GRCh38.p12 chr 3p13 3:178661712	C > A	A = 0.144	Intron
rs2861221	*CELF4*	GRCh38.p12 chr 18 18:37360216	C > G	G = 0.188	Intron
rs3020434	*ESR1*	GRCh38.p12 chr 6 6:152037805	C > T	T = 0.136	Intron
rs11742635	*FBN2*	GRCh38.p12 chr 5 5:128453101	G > T	T = 0.135	Intron
rs12457644	*CELF4*	GRCh38.p12 chr 18 18:37365013	G > A	A = 0.170	Intron
rs17677069	*FBN2*	GRCh38.p12 chr 5 5:128438445	A > G	G = 0.134	Intron

**Table 2 biomedicines-10-00508-t002:** Distribution of genotype frequencies in cases and controls in a large data set (255 samples). LM: all cases with leiomyomata. FP: cases with a familial predisposition. *p*-value less than 0.05 is shown in bold.

Gene	SNP ID	Genotype/Allele	Total	UL Patients without FP	UL Patients with FP	Control	*p*-Value Controls vs. All LM/ Controls vs. FP	OR (95%CI) All LM/FP
*CELF4*	rs2861221	CC/CG/GG	0.70/0.25/0.05	0.65/0.32/0.03	0.79/0.19/0.02	0.60/0.30/0.10	0.087/0.017	1.33/2.33
	rs12457644	GG/AG/AA	0.69/0.22/0.09	0.64/0.27/0.08	0.79/0.17/0.04	0.53/0.33/0.13	0.045/0.013	2.21/3.64
*FBN2*	rs11742635	GG/GT/TT	0.73/0.24/0.03	0.74/0.24/0.02	0.77/0.22/0.01	0.57/0.37/0.07	0.137/0.025	1.99/3.33
	rs17677069	AA/AG/GG	0.69/0.25/0.05	0.60/0.40/0	0.79/0.17/0.04	0.53/0.30/0.17	0.007/0.001	2.6/3.95
*KCNMB2*	rs12637801	CC/CA/AA	0.75/0.23/0.02	0.76/0.22/0.02	0.79/0.18/0.03	0.53/0.47/0	0.006/0.010	4.4/3.95
*ESR1*	rs3020434	CC/CT/TT	0.62/0.33/0.05	0.59/0.27/0.08	0.71/0.27/0.03	0.43/0.50/0.07	0.020/0.005	2.84/4.09

**Table 3 biomedicines-10-00508-t003:** Distribution of genotype frequencies in multiple and single myoma cases. A statistically significant difference is observed only in the case of rs12637801. The homozygous variant of the CC allele can be a predictor of the development of multiple uterine fibroids.

Gene	Genotype	Single Myoma, *n* = 81	Multiple Myoma, *n* = 134	Distribution of Alleles, χ^2^ Test, *p* Value
*rs3020434–ESR1*	CC	46 (56.7%)	87 (64.9%)	0.492
	CT	30 (37%)	40 (29.8%)	
	TT	5 (6%)	7 (5.2%)	
*rs11742635–FBN2*	GG	59 (73%)	98 (73.1%)	0.609
	GT	14 (24%)	33 (24.6%)	
	TT	2 (3%)	3 (2.2%)	
*rs124577644–CELF4*	AA	5 (6%)	3 (2%)	0.232
	AG	22 (26.8%)	27 (20%)	
	GG	54 (67%)	94 (70.1%)	
*rs12637801–KCWMB2*	AA	3 (4%)	1 (0.7%)	**0.031**
	AC	10 (12%)	33 (25%)	
	CC	67 (83.5%)	96 (71.6%)	
*rs2861221–CELF4*	CC	53 (65.6%)	96 (71.6%)	0.138
	CG	26 (32.8%)	31 (23%)	
	GG	1 (2%)	7 (5%)	
*rs17677069* *–FBN2*	AA AG GG	57 (70%) 13 (16%) 11 (14 5)	100 (74.6%) 25 (18.7%) 9 (6.7%)	0.238

## Data Availability

Not applicable.

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
