# Peer review of "Familial Predisposition to Leiomyomata: Searching for Protective Genetic Factors"

_biomedicines, 2022, doi:10.3390/biomedicines10020508_

Round 1

Reviewer 1 Report

The present study aims to identify genetic factors (polymorphism) associated with a lower risk of UL. The number of patients included is sufficient to answer the question by GWAS.

Of note, heritage of this benign condition exists, but this risk factor is relatively low in the Caucasian population (see epidemiological data). Therefore, larger cohort may be considered. The authors did not include patient of African origin who show higher lineage risk to develop symptomatic myomas.

I have a major concern about collecting data for family inheritance: because most myomas remain asymptomatic, there is a great risk of misclassification in subgroup 1B. Likewise, for subgroup1A, the patient's declaration of familial UL in a questionnaire (line 110) must also be medically documented. Any other gynecological pathology that the patient could confuse (STUMP, adenomyoma, polyp, etc.) in her heritage must be ruled out. Unless there is a clear explanation of the data collection, this impressive amount of work could be terribly biased. Please clarify this essential point.

Should this be clarified, I also have other concerns:
1. Additional statistical work should be added and other risk factors should be taken into account. Calculate the power of analysis and compute the ad-hoc sample size required. Perform binary logistic regression and adjust the results per age-women. Take into account biais (premature death of mother).

2. Statistical significance is not defined in the MM. Please adjust the p-value to the false discovery. In figure 1, the p-values ​​are expressed in -log10 and barely readable; please add p values ​​in the body of the text. Consider revising the legend for Figure 1.

3. The results of 3.1.1 and 3.1.2 should be presented in tables for ease of reading.

4. line 248: delete the instruction and give a title.

5. Table 2: indicate the proportions and underline the significant differences.

6. Line 302: Please discuss other somatic subtypes (ie HMGAs).

7. Writing can be improved (English grammar and conventions). Please be consistent in spelling, spaces and numbers.

Author Response

Dear reviewer, we are very grateful for the detailed analysis of our manuscript.

  1. Additional statistical work should be added and other risk factors should be taken into account. Calculate the power of analysis and compute the ad-hoc sample size required. Perform binary logistic regression and adjust the results per age-women. Take into account biais (premature death of mother).

We did not calculate the power of analysis for the whole-genome part of the study because we understand that the sample size is small and no SNP reached statistical significance, so we selected SNPs with the lowest p-values. As we divide our patients into two groups that strongly differ in their age and clinical history, we don't think that any adjustments will change these groups.

  1. Statistical significance is not defined in the MM. Please adjust the p-value to the false discovery. In figure 1, the p-values are expressed in -log10 and barely readable; please add p values in the body of the text. Consider revising the legend for Figure 1.

The significance level was adjusted using the Bonferroni correction. We added p values in the body of the text. We also made a revision of the legend for Figure 1.

  1. The results of 3.1.1 and 3.1.2 should be presented in tables for ease of reading.

In the first version of this manuscript, we entered the data results of 3.1and 3.2 in the tables. However, later we decided that these tables were superfluous in the article. We can list them in the Supplemental materials.

  1. line 248: delete the instruction and give a title.

Did you mean a line 432? Because in our file lie 248 is:

“of four of six studied SNPs (rs3020434, rs124577644, rs12637801, rs17677069) when compared”…

We have inserted the name of the Table 1.

  1. Table 2: indicate the proportions and underline the significant differences.

The proportions genotypes for each group are indicated in corresponded columns, significant differences are underlined.

  1. Line 302: Please discuss other somatic subtypes (ie HMGAs).

We focused our attention on fibroids with somatic mutations in the gene MED12, since these mutations are most often found in these tumors, especially when we study families with a burdened history. Most familial cases of the disease are associated with the development of multiple fibroids, whereas overexpression of the HMGA2 gene occurs most often in solitary fibroids.

  1. Writing can be improved (English grammar and conventions). Please be consistent in spelling, spaces and numbers.

We consulted with native English speakers to correct our text.

Despite the lower risk of fibroids in the Caucasian population (in comparison with others, for instance African population, the development of uterine fibroids among our patients is the most common reason for gynecological operations in our country. Our medical center is located in Russia, where there is almost no population of African descent. Such patients are treated with us no more than several times a year for all medical problems, not only about fibroids.

We use a questionnaire for collecting data for family inheritance, however we took into account data about operations of myomectomy or hormonal treatment of fibroids that were applied to mothers and other close relatives of patients. In addition, relatives of patients included in group 1А in most cases were also treated at our medical center and have medical documentation about the course of the disease. We also took into account the presence of fibroids according to ultrasound data obtained during the examination of patients’ relatives in our medical center. Thus, we had access to medical data not only of patients of group 1A but also of their relatives. This gave us the opportunity to exclude any other gynecological pathology that the patient could confuse (STUMP, adenomyoma, polyp, etc.).

Of course, we understand that some patients with a familial history to fibroids could be included not only in group 1A, but also in others due to the fact that fibroids in relatives were not diagnosed. However, this does not contradict the fact that group 1A managed to recruit the maximum number of patients with a hereditary predisposition to the disease.

Reviewer 2 Report

Well planned and performed study. I only wonder if Authors have data concerning the time of onset or recognition of fibroids in the study group. It would be interesting to perform analysis using such data.

Minor remarks:

1/ line 34 – instead of nodules more proper form is fibroids (alternatively tumors)

2/ line 40 – explain the abbreviation LM (last myomectomy?)

3/ line 107 – ovarian masses should be changed into benign ovarian tumors/ cysts

4/ line 174 – please replace “nodes” with fibroids (alternatively tumors)

5/ line 180 – “fibroid” is enough, next sentence – replace “nodules” with “fibroids” (please adopt these changes everywhere in the text)

Author Response

Dear reviewer, we are very grateful for the detailed analysis of our manuscript.

We have carefully corrected the text of our article in accordance with your comments.

Minor remarks corrected:

1/ line 34 – instead of nodules more proper form is fibroids (alternatively tumors)

Done

2/ line 40 – explain the abbreviation LM (last myomectomy?)

LM = Leiomyomata.

3/ line 107 – ovarian masses should be changed into benign ovarian tumors/ cysts

Done

4/ line 174 – please replace “nodes” with fibroids (alternatively tumors)

Done

5/ line 180 – “fibroid” is enough, next sentence – replace “nodules” with “fibroids” (please adopt these changes everywhere in the text)

Done

Round 2

Reviewer 1 Report

  1. [data source]: In their rebuttal, the authors clarified the main criticism of their study, stating that "the parents of the patients included in the 1А group have, in most cases, also been treated at our medical center and have medical documentation of the course of the disease." I thank the authors for this clarification. However, this key information should be made public; surprisingly, it is not included in the revised manuscript (should be present at least by line 79 in the revised document, v2).

  2. [variability in sampling]: As it is the first attempt to identify a historical lineage of SNP associated to uterine fibroids using genome-wide SNP-genotyping, and it may serve as a reference for future research, only certified history should be included (not "most", as stated in the rebuttal). Alternatively, this should be discussed accordingly, taking all the necessary precautions and considering the limitations. Similarly, the place of the asymptomatic fibroids should be added in the discussion. Please amend accordingly.

  3. On a personal basis : as the authors receive a chance to resubmit their work, I would take this as an opportunity to add limitations in the text (descriptive study, only correlations, no extrapolation to other ethic groups), should their data not be confirmed in the future.

The authors have addressed the minor points from round 1.

Author Response

  1. We included information about the method of a family history confirmation (see lines 81-83).
  2. Actually, our research is not the first attempt to identify a historical lineage of SNP associated to uterine fibroids, we just proposed a more correct approach for sampling. Asymptomatic fibroids could not be included in our study at all, since all the patients included in the analysis underwent surgical treatment. We also defined family history as the presence of surgical interventions in female patients' relatives. The presence in the family of several relatives with cases of multiple fibroids requiring surgical intervention and often leading to relapses allows us to identify a group of patients with a seminal history for whom inheritance of a predisposition to the disease may occur with a high degree of probability.
  3. We have made a link to our as yet unpublished data on various regions of Russia where we investigated the frequencies of some SNPs of interest to us among the population. The phrase was inserted into the text:
    Certainly, our results can be extrapolated only to the European population and only with some restrictions, since population frequencies can vary greatly, however, within the Russian population we found approximately the same frequencies of the studied alleles in samples collected in different regions (unpublished data).

Round 3

Reviewer 1 Report

All comments were addressed satisfactorily. The authors have responded to the main criticism, and included the related details in the manuscript.